# Human Factors Analysis of the Improved FRAM Method for Take-Off Quality Lateral Shift

Wenjian Ouyang [1,2,*] , Xusheng Gan [1,2], Yarong Wu [1,2], Kai Qu [1,2] and Jiabo Wang [3]

1  Air Traffic Control and Navigation College, Air Force Engineering University, Xi'an 710051, China; gxsh15934896556@163.com (X.G.)
2  National Key Laboratory of Air Traffic Collision Prevention, Xi'an 710051, China
3  China PLA 93886 Unit, Urumqi 830006, China
*  Correspondence: sxoywj@163.com

**Featured Application: The improved FRAM method can add a safety barrier to the feedback loop after functional resonance and verify the effectiveness of the safety barrier in a new round of flight data.**

**Abstract:** This article proposes an improved FRAM method based on the traditional FRAM method, using the study of aircraft take-off quality as an example to illustrate the operation method of the improved FRAM. To address the impact of pilot operations on take-off instructions during aircraft take-off, a functional network model was constructed based on the improved FRAM (functional resonance analysis method) method for the take-off and roll stages of the aircraft. On the basis of the aircraft take-off taxiing model, a simulation was used to sample the take-off data from the pilot many times under different conditions, and the data were put into the safety envelope for comparative analysis to find functional modules with abnormal changes. Using the functional network model, the resonance relationship between the abnormal module and other related functional modules was determined. According to the resonance relationship, setting up a safety barrier can reduce the risk of accidents. Finally, the safety barrier was substituted back into the improved FRAM method to verify the effectiveness of the safety barrier. Compared with the traditional FRAM method, the improved FRAM method can make full use of historical data, loop iteration, repeated verification, and continuous improvement until the final result reaches the user's expected goal. The improved FRAM method reduces the dependence on expert evaluation and experience, so its conclusions have higher objectivity and reference value.

**Keywords:** take-off running; FRAM; human factors; safety quality; functional barrier



## 1. Introduction

The take-off stage of an aircraft is the most accident-prone stage next to the landing stage. When the aircraft is granted a release permit, for the pilot, the environmental factors that affect takeoff are reaching their ability range standards. Therefore, the technical level of the pilot is another important standard for measuring the take-off quality of aircraft. The safety of the aircraft in the take-off phase is crucial to the safe operation of the whole flight. As important factors affecting the take-off quality in the take-off phase, human factors should be given sufficient attention. According to statistics, 65% of flight accidents occur in the take-off or landing phase [1]. From the perspective of the flight crew, the control of multiple navigation elements, such as speed, heading, braking, and altitude, during take-off has more stringent requirements and has a resonance response to multiple external environmental factors. Among the human factors that affect the take-off quality in the take-off stage, the pilot's status and operation are difficult to study and analyze through quantitative methods, while the commonly used qualitative analysis methods are highly dependent on subjective evaluations by analysts and experts and lack objectivity.

The FRAM model is a network framework established by the upstream and downstream relationships between system sub-modules [2]; the variation in one sub-module has an impact on the other sub-modules, which can lead to unsafe events that cause the functional resonance of the entire system. However, the traditional FRAM method only uses the flight data collected in the current single open-loop analysis, which is only effective for the analysis of this event, and the analysis results are not universal for similar events; the results of this analysis and experience cannot be fully applied to other similar scenarios. Therefore, an improved FRAM method is proposed in this paper, which sets up a corresponding safety barrier through the resonance relationship between the sub-modules of the system and replaces the output result with the improved FRAM cycle. The validity of the output results is verified by using historical data and the new iteration's results.

This study focused on the analysis of the influence of the pilot's control ability on the take-off quality by constructing a take-off quality analysis model. The process involved building different scenarios in the simulation environment, collecting the aircraft take-off data, and looking for the mutated function module through the relationship between the data and the safety envelope and then through the resonance effect of the correlation between the mutated module and the function module to determine the problems affecting the take-off quality and to establish safety barriers from the aspects of the system and human factors to reduce the take-off safety risk.

## 2. Traditional Systematic Analysis of Human Factors

Most research on aircraft take-off quality has been aimed at the impact of environmental factors on aircraft take-off performance data, with relatively little consideration of human factors. In the analysis of human factors, the flight quality is analyzed through flight data mining and expert experience; an example of such a method is the analytic hierarchy process [3], but this method is too dependent on expert experience and lacks objectivity. For the analysis of safety accidents, we are familiar with FTA (fault tree analysis), ETA (event tree analysis), and FMEA (failure mode and impact analysis). These methods are based on the construction of a map and framework to form an analysis network, followed by analyzing the causes and impacts of the accident and putting forward measures and suggestions that can be applied accordingly. However, in analyses based on ETA, FTA, FMEA [4–6], and other methods, the connections between sub-modules in the system branch cannot be effectively combined, so it is easy to ignore the impact of resonance between the sub-modules.

Other methods for the systematic analysis of human factors currently include the popular Human Factor Analysis and Classification System (HFACS), the Technique for Retrospective and Predictive Analysis of Cognitive Errors (TRACER), the Human Factors Investigation Tool (HFIT), the Cognitive Reliability and Error Analysis Method (CREAM), A Technique for Human Event Analysis (ATHEANA), Systems-Theoretic Process Analysis (STPA), etc. The HFACS method is logical and easy to implement. It can accurately identify important inducing patterns of aircraft collision accidents/accident symptoms and identify the influencing factors and correlations that often appear in the patterns. From the literature, ref. [7] put forward the human factors analysis framework of CAAC on the basis of HFACS. The analysis method of the TRACEr model can address the lack of thorough analysis in the cognitive field to some extent, but on the other hand, it also puts forward certain requirements for the analyst's knowledge level. The authors of [8] used the TRACEr cognitive framework as a basis to predict the probability of controller error and develop error prevention measures. HFIT can be applied to human factors analysis in many fields, but the revision of the model needs a large amount of work, which needs to be combined with practice to replace a large number of quantum elements and items. The threshold for its use is low, and its dependence on the knowledge structure of the analyst is weak. Through the screening of the expert's identification degree, more objective analysis conclusions can be formed, and more effective measures for the control of human factors can be implemented. The authors of [9] investigated human factors in the offshore oil and gas

industry by using HFIT and proved their significance in accident remediation. The CREAM method's traceability analysis framework can effectively extract accident chains and assist in identifying root causes. This quantitative analysis can make predictions for various types of human error behaviors, and Reference [10] used the second-generation CREAM method to quantitatively analyze the probability of accidents caused by human error. The ATHENA method needs accurate prior probability knowledge as support in the quantitative analysis part, and it is difficult to carry out a quantitative analysis in the absence of prior knowledge. The most commonly used method is to use expert knowledge to compensate for the lack of prior knowledge, but such analysis results may have some generality. After exploring the limitations of traditional human reliability analysis, Reference [11] discussed the advantages and disadvantages of the ATHENA method in terms of the impact of human behavior and performance on the results. Based on the STAMP (Systems-Theoretic Accident Modeling and Processes) model, STPA describes the operation of a system as a control model, and it can fully consider the influence of objective factors, human factors, and management factors on the system operation by fully mining the weak links in the system operation and the interactive relationship between the sub-parts; it is a widely used analytical method in the current engineering field. Reference [12] used the STAMP accident model and STPA analysis technology to analyze unsafe control behavior and proposed a reasonable aircraft-landing taxi-braking strategy from a quantitative analysis perspective.

The FRAM, the functional resonance accident model (as in Figure 1) [13], focuses more on the analysis of the resonance effects on the sub-modules of the system than the above methods; its creator, Hollnagel, believes that what causes accidents is essentially a mutation in the normal sub-modules of the system [14]. Based on the FRAM method, Rogier synthetically analyzed the function modules of humans, technology, and organization and studied the resonance effect among related modules on aviation safety accidents; finally, the corresponding safety barriers were formulated to reduce the risk of aviation safety accidents [15].

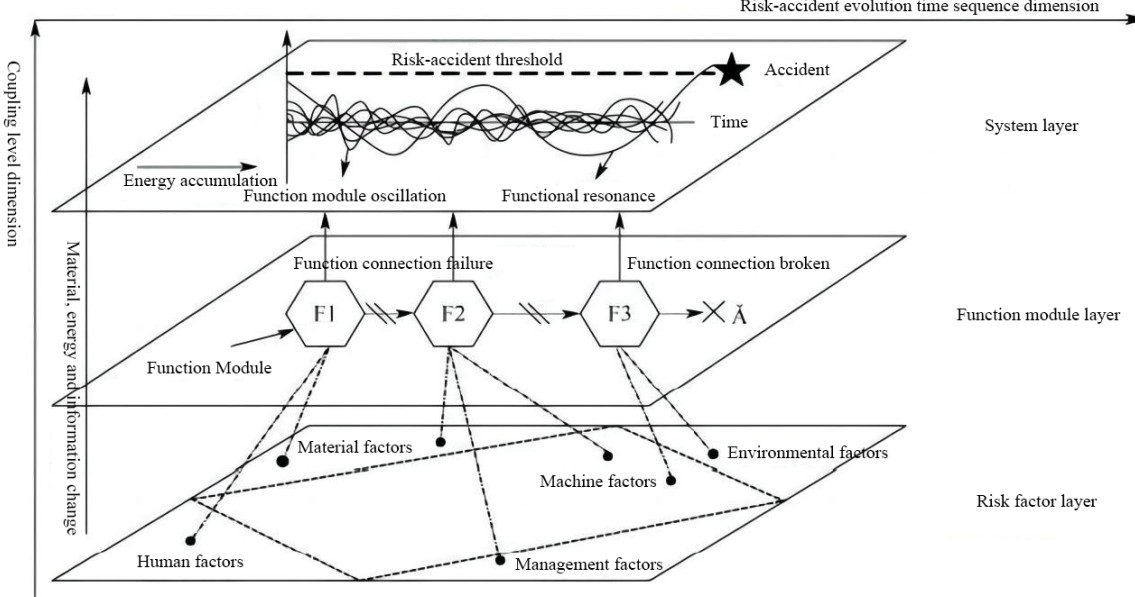

**Figure 1.** The FRAM functional resonance accident model.

The FRAM model has many applications in the field of safety assessment. Hollnagel, E. et al. used FRAM to analyze the crash of Comair Airlines (Delta Connection) Flight 5191 on 27 August 2006 in Lexington, Kenya, USA [16], and provided details not found in the NTSB analysis that identified risk factors resulting from the combination of functional modules, such as individuals, technologies, and organizations, that result in functional variation. Taking railway traffic supervision as an example, Belmonte, F. applied FRAM to the safety

analysis of complex social–technical systems [17] and proved that the FRAM model is different from the traditional safety analysis model, as individual and organizational competencies, as well as technical factors, can be integrated into a single model to enable collaboration between experts in different fields. Salihoglu, E. applied FRAM to the qualitative risk analysis of shipping operations to study environmental hazards caused by large oil spills [18], and it was demonstrated that FRAM can detect more detailed potential interactions among factors that cause accidents.

## 3. Improved FRAM Method Analysis Steps Are Introduced

The ETA, FTA, and FRAM methods analyze unsafe events through a network framework, while ETA starts the analysis from the initial events and finds the cause of unsafe events through induction and reasoning. The FTA method is based on the results of unsafe event analysis through the backward deduction of the causes of unsafe events. However, FRAM regards accident occurrence as resulting from the whole: as a type of systematic analysis method [19], it has both deductive reasoning logic and inductive reasoning. Therefore, the FRAM accident analysis model is more able to reflect the objective laws and accord with people's cognitive thinking.

The FRAM model consists of several functional modules (as in Figure 2), which make up the whole system. Each module has six nodes, which are:

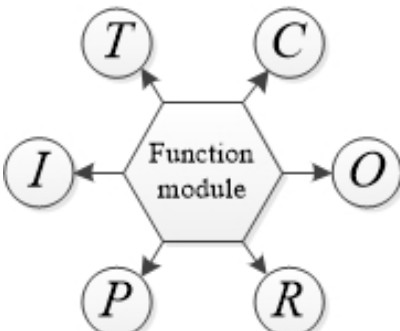

**Figure 2.** The FRAM functional module.

1. I (Inputs): function module of the input node;
2. P (Preconditions): conditions required for function module operation;
3. R (Resources): resources required for function implementation;
4. T (Time): time required for function execution;
5. C (Control): constraints on the execution of a function;
6. O (Outputs): function module output node.

The operational flow of the FRAM model is as follows. STEP 1: Identify and interpret functional modules; STEP 2: identify functional variability; STEP 3: focus on functional linkages and potential possibilities; STEP 4: manage and monitor functional variability.

However, when using the traditional FRAM model for analysis, only the accident-related data are collected before the analysis, and in the process of analysis, the data are used for a single open-loop analysis; the results of the analysis cannot rule out contingency and specificity, and the analysis method depends on the knowledge and experience of experts. Therefore, on the basis of the traditional FRAM model, the method in this paper iterates the safety barrier based on the analysis results in a new round of event analysis and cyclically collects the data to carry out the comparison, tests the resonance relation between the functional modules, and improves the use of historical data.

The improved FRAM process is shown in Figure 3.

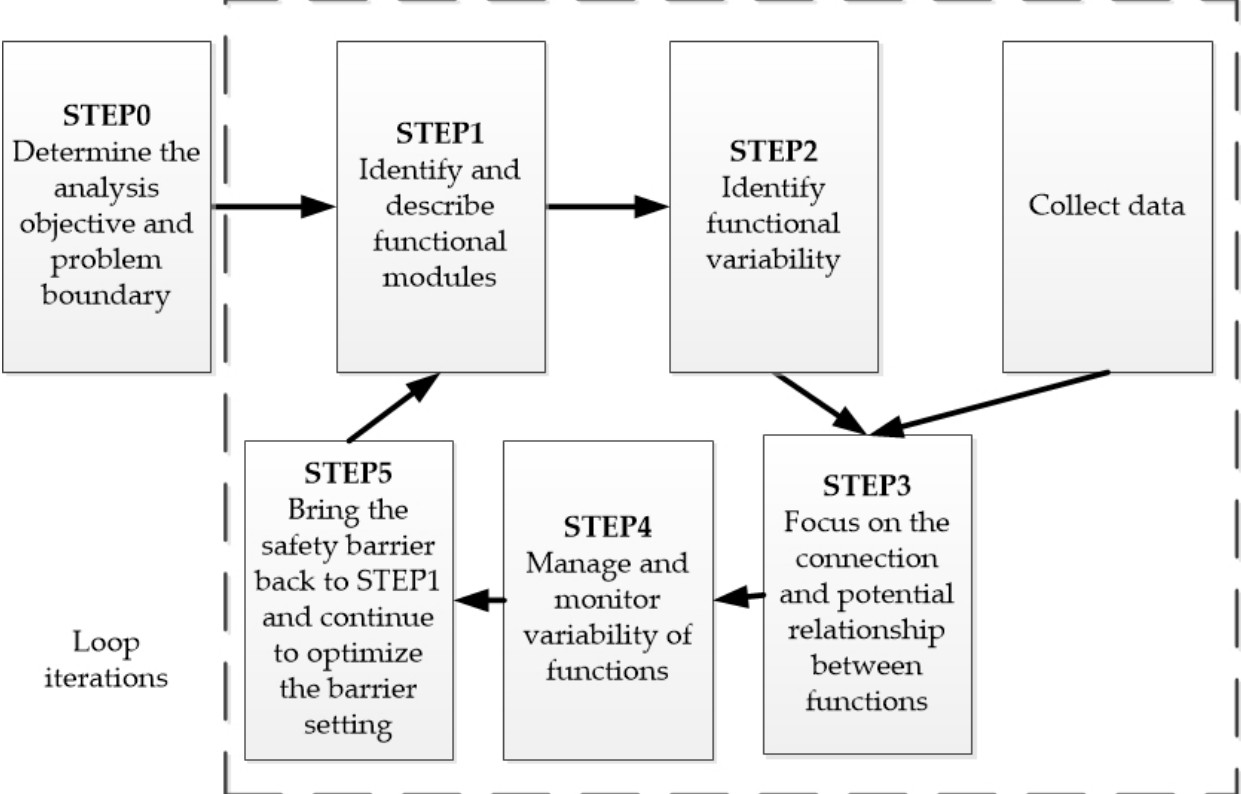

**Figure 3.** Improved FRAM analysis process.

**STEP 0:** Determine the objectives of the analysis and the boundaries of the problem, define the objectives and directions of the study, define the boundaries of the study, and prevent the boundaries from being unclear or too large; as a result, the number of functional modules and influencing factors related to the research problem increases geometrically, resulting in the problem of the coverage being too wide to obtain an effective solution.

**STEP 1:** Identify and describe functional modules; based on the hexagonal graph model of the FRAM function module, the system sub-modules are described and analyzed according to the (I) input node, (P) operation conditions, (R) resources, (T) time, (C) constraint conditions, and (O) output node, and the function of each sub-module is defined.

**STEP 2:** Identify the functional variability, clarify the relationship between sub-modules of the system, and determine the upstream module mutation and the impact of the downstream module. It should be noted that changes in environmental conditions have a certain relevance to the variation in each functional module. Therefore, after clarifying the upstream and downstream relationships of each functional module, the input and output data from the functional relationships or trends can be determined.

**STEP 3:** Pay attention to the connection and potential relationship between functions, understand the resonance relationship between all function modules through data mining, and consider the contingency of low-probability events. A lack of sample data may lead to missed or mistaken judgments of resonance relations. Therefore, focusing on the connections and potential relationships between functions requires a large amount of sample data.

**STEP 4:** Manage and monitor functional variability, and manage and detect functional module variability; with the aim of understanding the resonance relationship that occurs when functional modules mutate and discovering the modules associated with functional modules when unsafe events occur, a safety barrier is set between the mutated module and its associated module to prevent the occurrence of unsafe events. There are four kinds of safety barriers in common use: physical barriers, functional barriers, symbolic barriers, and invisible barriers.

**STEP 5:** Jump back to Step 1 for a new round of analysis. The safety barrier in Step 4 is substituted into FRAM, and the new output data are compared with the historical data to test the effectiveness of the safety barrier obtained in the previous round. Finally, examine the actual need to consider economic, time, human, and other practical cost problems to optimize the safety barrier program.

## 4. Take-Off Model Construction

The evaluation of aircraft take-off quality examines the following main points: lateral offset, airspeed above the ground, climb rate, pitch angle, flap position, etc. [20–22]. In this paper, the lateral offset, an important take-off quality measure, is analyzed. Take the common three-point aircraft take-off taxiing as an example: During the take-off process, once the tower provides the take-off command to the pilot, the pilot brakes and pushes the throttle to increase the engine speed. The aircraft take-off phase is shown in Figure 4. When the speed reaches a certain value, the pilot releases the brake so that the aircraft gains longitudinal acceleration, putting the aircraft into a three-point taxiing acceleration phase. When the aircraft taxiing speed reaches $V_R$ (speed to lift the front wheel), the pilot operates the aircraft to lift the front wheel in a two-point taxiing phase. When the aircraft speed is over $V_{lof}$ (ground speed), the aircraft, due to $L$ (lift), overcomes its own $G$ (gravity) and undergoes vertical acceleration into the climb phase. The take-off process is completed when the aircraft climbs to a safe altitude and retracts the landing gear. Therefore, the aircraft take-off can be roughly divided into two stages, ground taxiing and air climbing, as shown below.

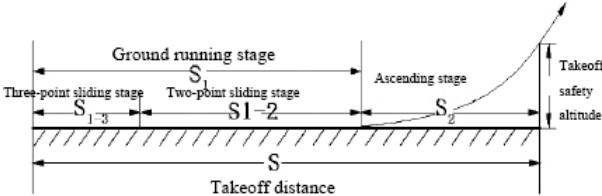

**Figure 4.** Schematic diagram of the take-off stage.

Based on the principle of dynamics, the calculation model of the aircraft taxiing distance is [23]:

$$S = \frac{1}{g} \int_0^{1.15 V_s} \frac{V \, dV}{\frac{P \cos \phi_P}{G} - f - \frac{\rho V^2 A}{2G}(c_x - f c_y)}$$

$S$ is the take-off taxiing distance, $P$ is the thrust of the engine, $V_S$ is the stall speed, $V$ is the taxiing speed of the aircraft, $\varphi_p$ is the angle between the thrust line of the engine and the baseline of the angle of attack, $C_x$ is the drag coefficient, $C_y$ is the lift coefficient, $G$ is the weight of the aircraft, $\rho$ is the atmospheric density, $A$ is the wing reference area, $g$ is the gravitational acceleration, and $f$ is the sliding friction coefficient. Consider the effect of wind direction and speed on take-off taxiing:

$$\begin{cases} V_x = V_T + V_{W_x} \\ V_y = V_{W_y} \end{cases}$$

$V$ is the taxiing speed, $V_T$ is the airspeed, $V_W$ is the wind speed, and $x$ and $y$ represent the longitudinal axis direction and transverse axis direction, respectively.

Thus, the offset of the aircraft on the $y$-axis can be expressed as:

$$y_s = V_y \cdot t$$

*t* is the taxiing time. In summary, the mathematical model of the taxiing path of the reaction of the aircraft can be summarized as follows:

$$
\begin{cases}
S = \dfrac{1}{g} \displaystyle\int_0^{1.15Vs} \dfrac{(V_T + V_{W_x})\,\mathrm{d}V_T}{\dfrac{P\cos\phi_P}{G} - f - \dfrac{\rho\,(V_T + V_{W_x})^2 A}{2G}(c_x - fc_y)} \\
y_s = V_y \cdot t
\end{cases}
$$

## 5. Take-Off Safety Analysis Based on Improved FRAM Method

Based on the analysis flow of the improved FRAM method, the take-off quality of pilots is analyzed. The improved FRAM method analysis process is shown in Figure 5.

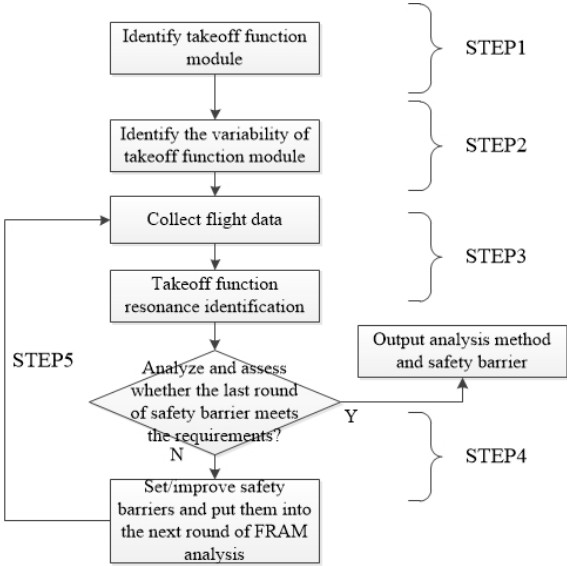

**Figure 5.** Take-off safety analysis based on the improved FRAM method.

**STEP 0: Identify the analysis goals and problem boundaries.**

The aim of the analysis is to determine the quality of the take-off by the pilot based on the distance of the lateral offset, the amplitude of the maximum lateral offset, the frequency of lateral offset correction, the airspeed above the ground, the climb rate, the pitch angle, etc. When the evaluation index exceeds the take-off safety envelope, the function module related to it will change. The ultimate goal of the analysis is to find the reason for the resonance between the variant module and the associated module and to prevent the resonance by setting a safety barrier so as to reduce the safety risk. The boundary of the problem is to take off after receiving the order and rise to a safe altitude after take-off.

**STEP 1: Identify and describe functional modules.**

During the take-off phase, the pilot begins the scheduled take-off procedure according to relevant regulations. From the take-off command, the pilot first pushes the brake down to increase the engine thrust, and when the engine reaches the required speed, then the brake is released to start taxiing the aircraft. During the taxiing process, due to the influence of human factors, machinery, and the environment, the thrust direction of the engine and the taxiing direction may be at a cross-angle, which makes it impossible for the aircraft to take off in a straight line; at this time, the pilot attempts to correct the direction of the aircraft through road signs and other visual information so that the aircraft can take off safely within a moderate range of deviation. After leaving the ground, the aircraft ascends to a safe altitude according to the planned take-off method while ensuring the lift, altitude, and heading of the aircraft are correct to complete the entire take-off phase.

During the take-off process, each relevant function module is connected with the others; through their interaction, the function modules can be input, and relations among them are the outputs, which can also be mutual resources or constraints. According to the

pilot's control flow during take-off, the functional modules are divided into three parts, as shown in Table 1. The functional modules can be classified according to human operation (H), technology (T), mechanical factors (M), and organizational management (O).

**Table 1.** Pilot function modules.

| Serial Number | Module | Category | Serial Number | Module | Category |
|---|---|---|---|---|---|
| F1 | Throttle | H, M | F5 | Program operation | O |
| F2 | Brake | H, M | F6 | Dynamic disposal | H |
| F3 | Flaps, elevators | H, M | F7 | Environment condition | T, O |
| F4 | Rudder, steering wheel | H, M | F8 | Lift-off | H |

F1, F2, F3, and F4 are directly operated by the pilot. F5, F6, and F7 are the influencing factors of F1, F2, F3, and F4. The output of F8 is directly related to the quality and safety of the take-off. The function and structure of F1, F3, F4, and F8 are analyzed in detail in Tables 2–5.

**Table 2.** The "F1: Throttle" function structure.

| Functional Unit | Performance |
|---|---|
| Input (I) | Throttle size |
| Output (O) | Acceleration |
| Resources (R) | Throttle lever |
| Time (T) | The whole process of take-off |
| Constraint (C) | 0–100% throttle |
| Premise (P) | The take-off speed did not reach the ground speed |

**Table 3.** "F3, flap, elevator" functional structure.

| Functional Unit | Performance |
|---|---|
| Input (I) | Flap and elevator position |
| Output (O) | Lift and vertical acceleration |
| Resources (R) | Operating procedures |
| Time (T) | The whole process of take-off |
| Constraint (C) | Required lift |
| Premise (P) | Requirement of flight procedure and insufficient lift |

**Table 4.** "F4: Steering device" functional structure.

| Functional Unit | Performance |
|---|---|
| Input (I) | Side deviation, course deviation angle, and program operation |
| Output (O) | Course and turn rate |
| Resources (R) | Rudder and steering wheel |
| Time (T) | Start taxiing to take off and climb to safe altitude |
| Constraint (C) | Included angle of track |
| Premise (P) | Lateral displacement during take-off |

**Table 5.** "F8: Off-Ground" functional structure.

| Functional Unit | Performance |
|---|---|
| Input (I) | Airspeed, climb rate, pitch angle, and rudder |
| Output (O) | Aircraft lift-off attitude |
| Resources (R) | Airspeed meter, gyroscope, and pavement marks |
| Time (T) | The wheel is off the ground at a safe height |
| Constraint (C) | Airspeed, lift, and position |
| Premise (P) | The aircraft has reached the lift-off speed |

According to the structure of the above function modules, each function module in the take-off phase can be connected to the others to form a functional network, As shown in Figure 6, where the solid lines represent the relationships between the pilot's direct operation modules, and the dashed lines represent the relationships between the impact factor modules.

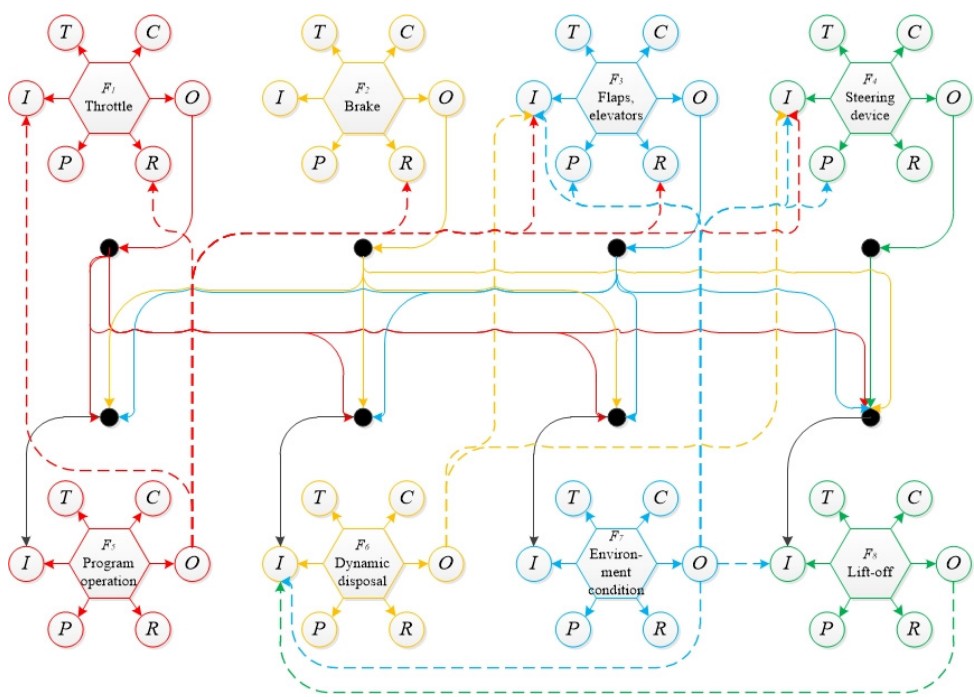

**Figure 6.** Take-off stage functional network.

**STEP 2: Identify functional variability.**

From the function module, it can be seen that the pilot's operation directly affects the take-off quality, and differences in the aircraft in F3 and the pilot's operation habits affect the use of flaps and elevators and further affect the time when the pilot lifts the front wheel; finally, the position, airspeed, and rate of ascent of the aircraft during take-off and ground departure are different.

Assume a take-off speed of 275–300 km/h, a lift coefficient of 0.89, a drag coefficient of 153.20, a runway slip friction coefficient of 0.005, an average thrust of 12,500 × 2 kgf, and a wingspan of 34 m². The take-off angle of attack is 10°, the take-off weight is 23 tons, the airfield pressure is 1013.25 hpa, and there is no wind. According to the above dynamic taxiing calculation formula, the take-off taxiing distance of the aircraft is 450–650 m. According to the Chinese military standard GJB34A-2012, the velocity v2 Min is $\geq$ 1.15 VS at the safe altitude. If the aircraft take-off taxiing distance is too short, it may cause the aircraft to leave the ground with insufficient speed, and there is a risk of stalling; if the taxiing acceleration distance is too long, it may cause the aircraft to take off below the minimum safe altitude, and there is a risk of hitting a ground obstacle. Considering the relationship between the ground velocity and the ground position, suppose that the airport is at an altitude of 0 m and the runway length is 2600 m to ensure that the aircraft can be raised to a safe altitude of more than 20 m at the end of the runway and that the pilot operates the aircraft in accordance with the standards of normal taxiing acceleration. The evaluation criteria are shown in Table 6.

**Table 6.** Range of throttle function changes.

| Take-Off Position (m) | Function Changes | Safety Quality |
| --- | --- | --- |
| dx < 450 | The running distance is too short, and there is a risk of stalling | Unqualified |
| 450 ≤ dx < 500 | The distance is slightly too short, but within the boundary | Qualified |
| 500 ≤ dx < 650 | Good control of running distance | Good |
| 650 ≤ dx < 1600 | The distance is slightly too long, but within the boundary | Qualified |
| dx > 1600 | The running distance is too long, and the ground clearance may be lower than the safe height | Unqualified |

The operation of the rudder and steering wheel in F4 is a method to correct the lateral displacement of an aircraft by visual judgment when the aircraft produces lateral displacement during take-off along the runway midline. The F4 function module is designed to ensure that the flight path of the aircraft during take-off is as straight as possible with the center line of the runway. The rudder function module directly affects the horizontal distance of the horizontal axis (Y-axis-RRB), and the weight of the take-off phase is mainly measured by the lateral offset distance of taxiing, the maximum lateral offset amplitude of taxiing, and the frequency of lateral displacement correction; thus, the horizontal distance along the transverse axis is used to evaluate the quality of rudder control, as shown in Tables 7–9.

**Table 7.** Function changes in "F2: rudder" (1).

| Lateral Displacement Distance (m) | Function Changes | Safety Quality |
| --- | --- | --- |
| 2 ≥ dy > 0 | Good steering control | Good |
| 6 ≥ dy > 2 | There is a risk of deviation from the runway, but within range | Qualified |
| dy > 6 | The risk of deviation is high | Unqualified |

**Table 8.** Functional changes in "F2: rudder" (2).

| Maximum Lateral Displacement Distance (m) | Function Changes | Safety Quality |
| --- | --- | --- |
| 3 ≥ dy > 0 | Good steering control | Good |
| 7 ≥ dy > 3 | There is a risk of deviation from the runway, but within range | Qualified |
| dy > 7 | The risk of deviation is high | Unqualified |

**Table 9.** Function changes in "F2: rudder" (3).

| Lateral Displacement Correction Frequency | Function Changes | Safety Quality |
| --- | --- | --- |
| 2 ≥ n > 0 | Good steering control | Good |
| 4 ≥ n > 2 | Normal direction operation, within the range | Qualified |
| n > 4 | Poor direction control and poor flying skills | Unqualified |

The take-off taxiing distance of a certain aircraft is 450−650 m, according to the safety envelope of the off-ground position, the lateral displacement distance, and the lateral displacement correction frequency. If the relevant parameters of the aircraft take-off

position and lateral displacement break through the safety envelope, the corresponding upstream module releases the mutation. Then, we check the module of the upstream phase by using video playback or flight parameter interpretation. The security envelope is shown in Figure 7.

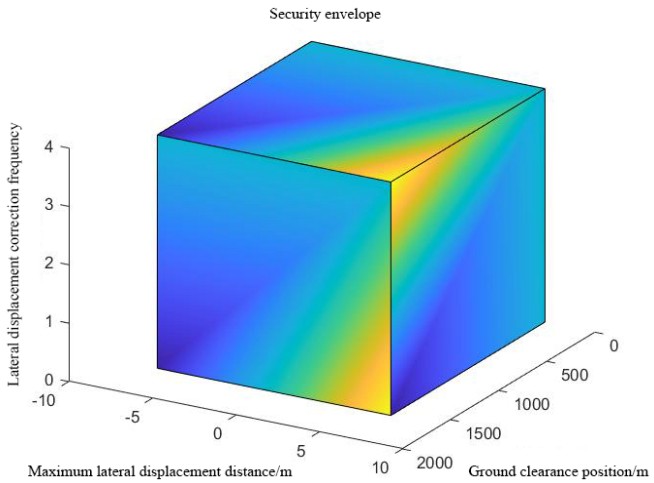

**Figure 7.** Security envelope.

**STEP 3: Focus on the connections and potential relationships between functions.**

In order to accurately identify the resonance relationship between all functional modules, the aircraft take-off process was analyzed using simulation tests, and the take-off model was constructed. The parameters of the take-off taxiing-phase model based on the take-off taxiing model and the disturbance factor are shown in Table 10.

**Table 10.** Take-off model parameter settings.

| Name/Variable | Value | Unit |
| --- | :---: | :---: |
| Lateral displacement distance dy | N (0,7) | m |
| Transverse maximum displacement distance DX | N (0,10) | m |
| Lateral displacement frequency | N (0,8) | / |
| Brake release time | 0 | s |
| Airspeed | N (0, n) | s |
| Wind direction | N (−90,90) | ° |
| Wind speed | N (0,10) | m/s |
| Take-off climb gradient | 3% | m/s |
| Actual airport pressure | 1013.25 | hPa |
| Aircraft model | / | / |
| Take-off weight | 23 | t |
| Maximum thrust | 2 × 12,500 | kgf |
| Wingspan area | 34 | m$^2$ |
| Departure point | (400, 2000) | m |
| Runway width | 45 | m |
| Runway friction coefficient | 0.005 | / |
| Lift coefficient | 0.89 | / |
| Resistance coefficient | 153.2 | / |

In the study of lateral offset, Reference [24] used the Kolmogorov–Smirnov test to analyze the distribution model of the aircraft wheel trajectory's lateral offset data, and it was verified that the lateral distribution of the observed model's wheel trajectory is more in line with a skewed distribution [25], as shown in Figure 8.

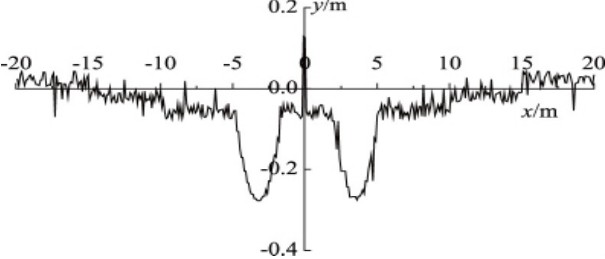

**Figure 8.** Measured transverse wheel track curve after signal processing.

By referring to the lateral offset fitting model and using the skewness distribution to simulate take-off, the main reference data, such as the lateral offset distance, the maximum lateral offset amplitude of taxiing, the lateral displacement correction frequency, airspeed above the ground, the rate of climb and the pitch angle, are obtained. The take-off quality analysis for the lateral offset mainly focuses on the lateral offset distance, maximum lateral offset amplitude, and lateral displacement correction frequency. The simulated target pilot P1 and skilled pilot P2 took off 10 times at the same time and in the same environment, as shown in Tables 11 and 12.

**Table 11.** Horizontal offset simulation data.

| Serial Number | Departure Point | Maximum Lateral Displacement Distance | Maximum Lateral Displacement Distance | Lateral Displacement Correction Frequency |
|---|---|---|---|---|
| 1 | 754 | 1.68 | 8.83 | 1 |
| 2 | 447 | 2.03 | 2.34 | 6 |
| 3 | 646 | 6.98 | 6.98 | 2 |
| 4 | 1118 | 1.86 | 6.03 | 4 |
| 5 | 587 | 3.37 | 9.01 | 2 |
| 6 | 1256 | 0.54 | 4.20 | 4 |
| 7 | 828 | 1.46 | 1.75 | 1 |
| 8 | 650 | 0 | 0 | 0 |
| 9 | 811 | 0.86 | 0.91 | 2 |
| 10 | 1868 | 2.37 | 2.54 | 1 |
| Skewness | 1.3097 | 1.5807 | 0.2826 | 0.7974 |
| Mean value | 896.5000 | 2.1150 | 4.2590 | 2.3000 |
| Standard deviation | 418.4331 | 1.9857 | 3.2710 | 1.8288 |

**Table 12.** Horizontal offset simulation data.

| Serial Number | Departure Point | Maximum Lateral Displacement Distance | Maximum Lateral Displacement Distance | Lateral Displacement Correction Frequency |
|---|---|---|---|---|
| 1 | 965 | 3.9531 | 3.9531 | 1 |
| 2 | 570 | 3.7179 | 3.7179 | 0 |
| 3 | 776 | 1.6381 | 2.3789 | 1 |
| 4 | 761 | 0.0014 | 3.0737 | 2 |
| 5 | 864 | 2.1635 | 4.9559 | 0 |
| 6 | 613 | 0.8309 | 1.0185 | 1 |
| 7 | 887 | 0.8771 | 2.7462 | 1 |
| 8 | 570 | 1.3032 | 3.3793 | 1 |
| 9 | 1638 | 0.3838 | 1.2447 | 2 |
| 10 | 532 | 2.9901 | 2.3789 | 2 |
| Skewness | 1.6776 | 0.4008 | −0.0241 | −0.1399 |
| Mean value | 817.6000 | 1.7859 | 2.8847 | 1.1000 |
| Standard deviation | 325.3590 | 1.3810 | 1.2069 | 0.7379 |

The take-off taxiing track of serial No. 7 is shown in Figure 9.

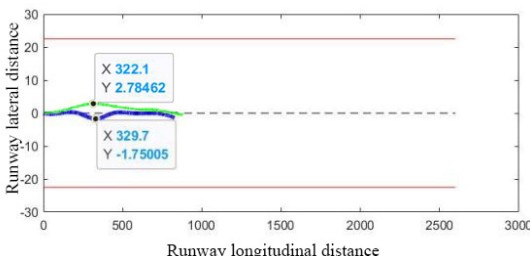

**Figure 9.** No. 7 take-off slip path.

In the path map, the middle part of the two solid red lines is the runway, the black dashed line is the centerline of the runway, the blue solid line is the take-off and taxiing path of the P1 aircraft, and the green solid line is the take-off and taxiing path of the P2 aircraft. The position where the longitudinal distance of the runway is 0 is the take-off line, and the direction of the take-off runway is 90°.

The data in Tables 11 and 12 were inserted into the security envelope, as shown in Figures 10–12.

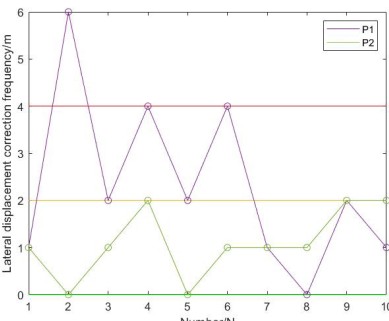

**Figure 10.** Lateral displacement correction frequency.

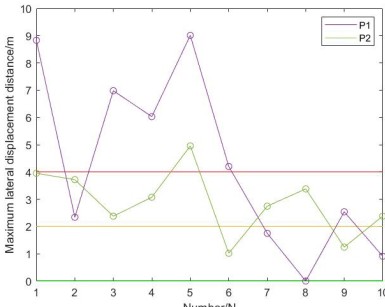

**Figure 11.** Maximum lateral displacement distance.

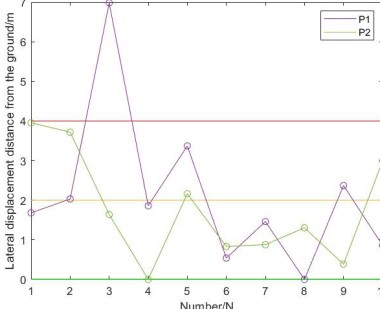

**Figure 12.** Lateral displacement distance from the ground.

The integration of the above data into the security envelope is shown in Figure 13.

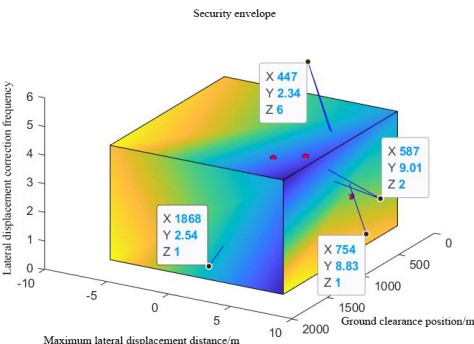

**Figure 13.** Safety envelope breakthrough.

In the horizontal comparison of P1 and P2 data, the data for P2 are generally better, indicating that the take-off quality is less affected by environmental factors during this period, and there is a certain gap in the operational skills of the P1 and P2 pilots.

A separate analysis was conducted on the take-off quality of the P1 pilots. Based on the data in the above figure, it can be seen that there are significant anomalies in the take-off simulation of P1 numbers 1, 2, 3, 5, and 10, and there is a significant anomaly in the flight quality. The maximum lateral offset distance is 9 m during the take-off with Nos. 1 and 5, and there is a huge risk of running off the runway, according to the above data from the "F4: Steering Device" function module analysis. The steering devices used by the pilot are the rudder and the steering wheel for airplane deflection and the turning control operation device; the function modules of "Side offset, course offset" and "Pilot's program operation, dynamic handling, and environmental conditions" are used as inputs and resources, respectively. The "Take-off process has lateral displacement" is used as the premise, and then the output of the module is "Heading, turning rate". Therefore, it can be inferred that when the pilot perceives that the lateral deviation between the plane and the center line of the runway exceeds the expected range, the pilot uses the "F4: Steering Gear" function module because of variations in "F6: dynamic disposal" or "F7: environmental conditions" (wind, surface pollutants, etc.); the lateral displacement distance of the aircraft is beyond the safety envelope, which means that the pilot has not corrected the lateral displacement well. Therefore, functional resonance is formed between F4, F6, and F7.

From the point of view of the take-off taxiing length, the serial No. 10 taxiing distance is too long to break through the safety envelope. From the data, we can see that the relative indexes of the lateral migration of serial No. 10 are in a good range, but the taxiing acceleration is relatively low, which causes the aircraft to require more time to accelerate to the take-off speed. The main factors that affect the take-off acceleration are the throttle and flap. There may be variations in the "F1: throttle" and "F3: Flap, elevator" function modules without considering the downwind take-off. During take-off, the pilot does not push the throttle to the end according to the procedure, and the flaps are placed in the prescribed position for the procedure; the acceleration of the aircraft does not reach the expected value because of the premature lifting of the front wheel by the pilot, which makes the take-off taxiing distance break through the safety envelope, and there is a high risk of running off the runway. Thus, functional resonance is formed between F1 and F5, between F3 and F5, and between F1 and F3 and F6.

In the takeoff environment of number 2, the wind direction is 30° wind at 7 m/s. In the case of strong crosswind, the difficulty of correcting the aircraft's direction is greater than when the crosswind is small. Among the 10 simulations, the frequency of lateral correction by the pilot is too high in only one operation, so it can be concluded that the ability of the pilots to correct lateral displacement is relatively good, and the main reason for breaking through the safety envelope is the result of the interaction between environmental

conditions and dynamic disposal. Thus, the variation in the functional module "F4: Steering Gear" in serial No. 2 forms a functional resonance with F6 and F7.

**STEP 4: Manage and monitor variability in functions.**

Based on the resonance relationship in Step 3, the reasons for the breach of the safety envelope are summarized in Table 13.

**Table 13.** Cause analysis of functional resonance.

| Resonance Relation | Cause Description |
|---|---|
| F4~F6~F7 | The influence of environmental factors is high, the pilots are not sensitive to lateral displacement, and the control of the direction of the aircraft is not sufficient to correct the deviation in time. |
| F1~F5 | Take-off does not follow procedures for the throttle, so it is not pushed to the maximum position, resulting in less thrust. |
| F3~F5 | The flap is not put in the designated position according to the procedure, the angle is too large, and the take-off windward resistance is too high. |
| F1, F3~F6 | The pilot's perception of the speed and acceleration of the aircraft is not accurate enough to be corrected in time. |

In terms of the resonance frequency of the functional modules, the frequencies of "F4: steering gear" and "F6: Dynamic Disposal" are the highest: "F6: dynamic disposal" means that, in the case of small differences in environmental conditions, the main reason for the lateral deviation of the aircraft that causes it to break through the safety envelope is that the technical level of the pilot's steering device is poor. To reduce the security risk, F4 and F6 should be placed on the safety barrier. Barriers are applied as shown in Tables 14 and 15.

**Table 14.** "F4: Steering device" safety barrier.

| Function Module Name | Type of Barrier | Description of Barrier Measures |
|---|---|---|
| F4 Steering device | Physical barriers | Strengthen the detection of environmental data, implement the cleaning of No. 1 pavement, strengthen the management of unit resources, enable them to supervise and inspect each other, and ensure necessary reminders are provided by control and command personnel. |
| | Functional barriers | Strengthen the regular repair and inspection of installed equipment, which should be checked by the pilot before take-off. |
| | Symbolic barrier | Make good use of pavement markings, satellite positioning, and radio navigation equipment to strengthen the monitoring of the aircraft position and attitude. |
| | Invisible barrier | Strengthen the day-to-day theory and simulator training studies, summarize the flight experience, and enhance the flight technique of the pilot. |

**Table 15.** "F6: Dynamic Disposal" security barrier.

| Function Module Name | Type of Barrier | Description of The Barrier Measures |
|---|---|---|
| F6 Dynamic disposal | Physical barriers | Items should be checked with a checklist in places where it is easy to forget to forget warning signs, etc. |
| | Functional barriers | Tower-related personnel should strengthen dynamic monitoring and alerts. |
| | Symbolic barrier | Crew members should repeat instructions and check each other's consistency in operations. |
| | Invisible barrier | Unit resource allocation, attention allocation, and situational awareness should be strengthened. |

**STEP 5: Conduct a new round of validation.**

Referring to Tables 14 and 15, before the P1 pilots embark on their next flight, a plan to improve the take-off operation quality of P1 pilots based on F4 intangible barriers and F6 functional barriers, symbolic barriers, and intangible barriers should be designed: Plan 1: Simulate flight operation training; Plan 2: Ensure controller's dynamic observation and reminders; Plan 3: Have experienced pilots impart their experience. Other plans are based on F4 physical barriers, functional barriers, and symbolic barriers and F6 physical barriers: Plan 4: Maintain runway signs, lighting, and other equipment; Plan 5: Strengthen environmental cleaning. The data shown in Table 16 are from simulations assuming that pilot P1 has undergone the improvements in Schemes 1, 2, and 3 and performs a take-off simulation under the same environmental conditions.

**Table 16.** P1 take-off simulation data in round 2.

| Serial Number | Departure Point | Maximum Lateral Displacement Distance | Maximum Lateral Displacement Distance | Lateral Displacement Correction Frequency |
|---|---|---|---|---|
| 1 | 680 | 3.4042 | 6.1970 | 3 |
| 2 | 806 | 4.0462 | 1.4762 | 3 |
| 3 | 499 | 4.0396 | 0.8850 | 1 |
| 4 | 900 | 0.6911 | 2.5942 | 1 |
| 5 | 474 | 3.2477 | 3.4335 | 1 |
| 6 | 552 | 2.2659 | 7.6100 | 2 |
| 7 | 507 | 4.9182 | 3.0484 | 5 |
| 8 | 1757 | 2.3778 | 7.0779 | 1 |
| 9 | 868 | 1.0681 | 3.1938 | 1 |
| 10 | 1288 | 1.7626 | 8.3793 | 5 |
| Skewness | 1.2516 | 1.5807 | 0.2642 | 0.8209 |
| Mean value | 833.1000 | 2.1150 | 4.3895 | 2.2000 |
| Standard deviation | 410.4641 | 1.9857 | 2.6857 | 1.6193 |

The comparison between the obtained flight quality data and the historical data of the previous round for P1 (Table 11) shows that there is no significant improvement, indicating that the flight barriers related to Schemes 1, 2, and 3 have not worked. Therefore, in the next round of the analysis, safety barriers that have not worked should be removed, and Schemes 4 and 5, generated based on other safety barriers, should be used. If the flight quality obtained through Schemes 4 and 5 shows significant improvement compared to the historical data of the previous round, it indicates that the flight barriers related to Schemes 4 and 5 have played a role. Therefore, it can be concluded that the reason for the low rate of good take-off flight quality for P1 pilots is due to their poor ability to cope with unfavorable external environments.

In the practical application of the improved FRAM method, in order to further verify the effectiveness and practicability of the safety barrier, the next take-off quality was analyzed after the safety barrier was applied. By comparing the collected data with the historical data, we can confirm that the safety barrier set up in the last round is reasonable and effective. When checking whether the safety barrier that was set up has caused an abnormality in the new system function module, if a new exception is generated, the exception is added to the next round of the improved FRAM cycle until the take-off quality is acceptable. Finally, through a cost–benefit analysis, the improvement program is determined.

A comparison of the traditional FRAM approach and the improved FRAM approach is shown in Table 17.

**Table 17.** Comparison and analysis of the model performance.

| Difference | Traditional FRAM Method | This Paper Improves FRAM Method |
| --- | --- | --- |
| Analysis process | Open-loop single analysis process. | Closed-loop feedback flow, rolling analysis. |
| Analysis phase | Post-accident analysis. | Systematic analysis of the whole process. |
| Analytical criteria | Highly dependent on expert knowledge and experience. | More attention is paid to objective data when comparing the results of multiple rounds of cycle analysis. |
| Analyze data | Only use the accident-related data; data utilization rate is low. | Full association of a large number of historical data; high utilization of information. |
| Analysis efficiency | A single model is only suitable for this single security analysis. | The same problem built by the model through improved rolling analysis can be used multiple times. |
| Analysis conclusion | The effect of the safety barrier cannot be verified. | The effect of the safety barrier is verified through the feedback loop and can be continuously optimized. |

## 6. Conclusions

(1) The analysis of aircraft take-off quality can be carried out systematically using the FRAM method according to the functional modules operated by pilots. By constructing a take-off taxiing model, the pilot can simulate the possible taxiing of the aircraft during take-off. According to flight data such as the lateral offset distance, the lateral distance correction frequency, the taxiing speed, the acceleration, the taxiing distance, and the simulated airport environment, the human factors that may affect the flight quality during the flight are analyzed.

(2) The FRAM model is specific to specific functional modules, can track the specific operation of pilots at a certain time, can more intuitively point out the human factors that affect the take-off quality, and has good practicality in finding problems and setting safety barriers.

(3) Compared with the traditional FRAM method, the safety barrier set by the improved FRAM method after functional resonance can enter the feedback loop, and the effectiveness of the safety barrier set in the previous round can be verified in the new round of flight data. The analysis results of each round can be compared with historical data, which can more objectively reflect the common problems of multiple pilots in take-off operations and correct them.

(4) After reaching the analysis conclusion and setting up the safety barrier, the improved FRAM method can repeatedly modify and iterate the new scheme according to the cost–benefit analysis until it meets the requirements of the analysts. Compared with the traditional FRAM method, the improved FRAM method can only obtain a safety barrier that cannot be modified, which is more practical.

**Author Contributions:** Conceptualization, X.G. and Y.W.; methodology, Y.W.; software, K.Q.; validation, W.O. and K.Q.; formal analysis, W.O.; investigation, J.W.; resources, J.W.; data curation, J.W.; writing—original draft preparation, W.O.; writing—review and editing, W.O.; visualization, W.O.; supervision, X.G.; project administration, X.G.; funding acquisition, Y.W. All authors have read and agreed to the published version of the manuscript.

**Funding:** This research was funded by the National Natural Science Foundation of China: "Modeling and Sensitivity Analysis of Task Safety for a 'Man-Machine-Environment' Complex System from an Uncertain Perspective", grant number: 52074309.

**Institutional Review Board Statement:** Not applicable.

**Informed Consent Statement:** Not applicable.

**Data Availability Statement:** Data sharing not applicable.

**Conflicts of Interest:** The authors declare no conflict of interest.

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
