# Peer review of "Human Factors Analysis of the Improved FRAM Method for Take-Off Quality Lateral Shift"

_applsci, doi:10.3390/app13085216_

Round 1
Reviewer 1 Report
I agree that analysis of human factor is important subject. However, I observed following concerns in this paper.
1. This study focused on lateral-shift during take-off. The importance of this specific factor is not clear on safety. What about other errors during take off? (e.g. timing of pulling back the stick at Vr) I'm also afraid landing is more critical than take-off.
2. The parameters in the numarical simulation should be justified, especially the probability distributions and their parameters. Is Gaussian really appropriate?
3. The advantage of the proposed method over FRAM is questionable. The paper says in Table 15 that "Full Association of a large number of historical data..." and "More attention is paid to objective data..." among the several advantages. However, the aspect of data size is questionable because the number of simulation is only 10. Moreover each case is analyzed individually so that essentially the process is the same as FRAM. Also, the aspect of data objectiveness is also questionable because the presented simulation uses arbitrary parameters.
Author Response
Point 1: This study focused on lateral-shift during take-off. The importance of this specific factor is not clear on safety. What about other errors during take off? (e.g. timing of pulling back the stick at Vr) I'm also afraid landing is more critical than take-off.
Response 1: Thank you for your guidance, as the primary purpose of this article is to introduce the improved FRAM method, so use the lateral offset of the aircraft during take-off, to illustrate the flow, advantages and disadvantages of improved FRAM method. I know that there are many factors that affect the safety of aircraft. If we expend on them all and analyze them one by one, we may end up with a hundred-page analysis report, therefore, in this paper, we only use the improved FRAM method to analyze the pilot operation, environment and other related factors in the takeoff lateral migration.
Point 2: The parameters in the numarical simulation should be justified, especially the probability distributions and their parameters. Is Gaussian really appropriate?
Response 2: Based on the trajectory deviation law obtained from others' research during takeoff, I have changed the parameters to skewed distribution and provided references in the article.
Point 3: The advantage of the proposed method over FRAM is questionable. The paper says in Table 15 that "Full Association of a large number of historical data..." and "More attention is paid to objective data..." among the several advantages. However, the aspect of data size is questionable because the number of simulation is only 10. Moreover each case is analyzed individually so that essentially the process is the same as FRAM. Also, the aspect of data objectiveness is also questionable because the presented simulation uses arbitrary parameters.
Response 3: The application of a large amount of historical and objective data is a characteristic of the improved the FRAM method. The use of data is mainly reflected in horizontal and vertical comparisons, in order to identify the modules that cause functional resonance. I have added horizontal comparisons between pilots in terms of content, which indicates that the problem is likely due to the pilot's own operational level. Then, I compare the longitudinal data of the same pilot before and after two times, To confirm that the real cause of the pilot's problems is the imperfect environmental equipment factors, and the pilot's operational level and coping ability have a certain gap compared to experienced pilots, resulting in poor takeoff quality of the aircraft.

Reviewer 2 Report
Take-off and landing phase are most critical phases and therefore human factor should be given additional attention. In the article authors present the functional network model of the aircraft take-off taxiing stage using the improved FRAM method, with the aim of examining the influence of the pilot's operation on the take-off command during the aircraft take-off process. Their upgraded FRAM approach may fully utilize historical data, loop iteration, repeated verification, and continual development, until the end result achieves the user's anticipated aim, which are the main advantages compared to the conventional FRAM method use. Their improved FRAM method lessens reliance on expert judgment and expertise, increasing the objectivity and reference of its results.
The article is nicely structured, with the methodology, model construction and analysis clearly explained in sequential steps. However, in order to improve the quality of the article, it is recommended to the authors to:
- Authors should include section dedicated to literature review of related work that was already done in the field and increase the number of references (16 is too little)
- Change Figure 1 which is in bad quality
- Change Figure 3 which is not clear and is hard to read
- In line 230 – mechanical factor is in lowercase, and variable T is not explained
- Line 276 – Tables 8 and 9 have different format
- Table 10 – Wingspan area – m2 should be with upper index
- Authors should check format, style and spelling throughout the whole article
Author Response
Point 1: Authors should include section dedicated to literature review of related work that was already done in the field and increase the number of references (16 is too little)
Response 1: Thank you for your guidance, I have supplemented the other human factor methods and explained them in the introduction, in addition, in the relevant chapters of the simulation model, I have introduced the content of skew distribution based on other people's research, there are now 25 references.
Point 2: Change Figure 1 which is in bad quality
-Change Figure 3 which is not clear and is hard to read
-In line 230 – mechanical factor is in lowercase, and variable T is not explained
-Line 276 – Tables 8 and 9 have different format
-Table 10 – Wingspan area – m2 should be with upper index
-Authors should check format, style and spelling throughout the whole article
Response 2: The above format content has been corrected, and in the post-inspection found similar problems in other forms, and together with the completion of the correction.

Reviewer 3 Report
1. Figure 1 need rework. The resolution of this figure not correspond to the requirements.
2. Fig 3 unreadable.
3. Table 1. Under "category column", incorrect commas. please recheck
4. line 213. A space missing between word "step" and number 1. same remark for lines 245, 286, 353 and 364.
Author Response
Point 1: Format issues
- Figure 1 need rework. The resolution of this figure not correspond to the requirements.
- Fig 3 unreadable.
- Table 1. Under "category column", incorrect commas. please recheck
- line 213. A space missing between word "step" and number 1. same remark for lines 245, 286, 353 and 364.
Response 1:The above format content has been corrected, and in the post-inspection found similar problems in other forms, and together with the completion of the correction.

Round 2
Reviewer 2 Report
The article is well organized, with the methodology, model construction, and analysis clearly explained in sequential steps. However, it could include a separate section dedicated to related work and literature review.
Author Response
Point 1: It could include a separate section dedicated to related work and literature review.
Response 1:Thank you for your guidance. I have added a new chapter as requested and adjusted the structure of the article to make it easier to read.
